

# A hybrid multi-objective whale optimization algorithm for analyzing microarray data based on Apache Spark

Amr Mohamed AbdelAziz[1], Taysir Soliman[2], Kareem Kamal A. Ghany[1,3] and Adel Sewisy[2]

[1] Faculty of Computers and Artificial Intelligence, Beni-Suef University, Egypt
[2] Faculty of Computers and Information, Assiut University, Egypt
[3] College of Computing and Informatics, Saudi Electronic University, Riyadh, KSA

## ABSTRACT

A microarray is a revolutionary tool that generates vast volumes of data that describe the expression profiles of genes under investigation that can be qualified as Big Data. Hadoop and Spark are efficient frameworks, developed to store and analyze Big Data. Analyzing microarray data helps researchers to identify correlated genes. Clustering has been successfully applied to analyze microarray data by grouping genes with similar expression profiles into clusters. The complex nature of microarray data obligated clustering methods to employ multiple evaluation functions to ensure obtaining solutions with high quality. This transformed the clustering problem into a Multi-Objective Problem (MOP). A new and efficient hybrid Multi-Objective Whale Optimization Algorithm with Tabu Search (MOWOATS) was proposed to solve MOPs. In this article, MOWOATS is proposed to analyze massive microarray datasets. Three evaluation functions have been developed to ensure an effective assessment of solutions. MOWOATS has been adapted to run in parallel using Spark over Hadoop computing clusters. The quality of the generated solutions was evaluated based on different indices, such as Silhouette and Davies–Bouldin indices. The obtained clusters were very similar to the original classes. Regarding the scalability, the running time was inversely proportional to the number of computing nodes.

## INTRODUCTION

A microarray is a high throughput laboratory tool used to expose samples (genes, contigs, non-coding sequences) to different experimental conditions simultaneously (*Bolon-Canedo et al., 2014*). It generates data that describe the expression profiles of samples during experiments. Analyzing microarray data can help researchers to discover valuable information about samples, such as identifying correlated genes (*Liu, Cheng & Tseng, 2013*; *Chou et al., 2007*), predicting patient response to specific treatments (*Ban et al., 2011*), and identifying different classes of cancer (*Saber & Elloumi, 2015*). Clustering has been an important data analysis tool (*Jain, Murty & Flynn, 1999*). Multiple clustering methods were proposed to analyze microarray data (*McDowell et al., 2018*;

Corresponding author
Amr Mohamed AbdelAziz,
amraziz@fcis.bsu.edu.eg

*Chandra & Tripathi, 2019*). These methods group samples with similar expression profiles into clusters as a way to reveal hidden patterns of samples (*Jothi, Mohanty & Ojha, 2016*).

Clustering methods used a single objective function to evaluate the quality of generated clusters (*McDowell et al., 2018*; *Chandra & Tripathi, 2019*). To improve the quality of the obtained clusters, clustering methods used multiple objective functions to evaluate the generated clusters (*Mukhopadhyay, Maulik & Bandyopadhyay, 2015*; *Acharya & Saha, 2018*; *Parraga-Alava, Dorn & Inostroza-Ponta, 2018*). This converted the clustering problem into a Multi-Objective Problem (*MOP*).

Pareto Optimization (*PO*) is one of the main techniques used to solve Multi-Objective Problems (*MOPs*) (*Freitas, 2004*). It aims to optimize the whole objectives simultaneously. It generates a set of best solutions called "non-dominated" solutions (*Freitas, 2004*). *PO* can be used to identify the best solutions regarding multiple objective functions. To provide efficient coverage of the solution space, researchers paid attention to combine *PO* with Swarm Intelligence (*SI*) methods. The latter mimics the intelligent behavior of organisms that live into groups to provide an efficient coverage of the solution space (*Talbi, 2009*). Many *SI* methods have been proposed to cluster microarray data based on *PO* (*Paul, Shill & Kundu, 2016*; *Mandal & Mukhopadhyay, 2017*; *Zareizadeh et al., 2018*).

A new hybrid *SI* method based on hybridizing Whale Optimization Algorithm (*WOA*) with Tabu Search (*TS*) (*WOATS*) was proposed for data clustering (*Ghany et al., 2020*). *WOATS* was tested over multiple real-life datasets and it was able to obtain high quality clusters regarding both homogeneity among cluster members and separation among clusters. The new hybrid method was extended to be applied for *MOPs* (*MOWOATS*), which was proposed by *AbdelAziz et al. (2019)*. *MOWOATS* used the memory elements of *TS* to enhance both the intensification and diversification techniques of the basic *WOA*. Also, *MOWOATS* used *PO* to ensure that all objectives are optimized simultaneously. *MOWOATS* stored non-dominated solutions in an Elite List (*EL*) inspired by *TS*. *MOWOATS* incorporated the crossover operator to improve the diversity of solutions and to ensure faster convergence rates. Due to these enhancements, *MOWOATS* was able to find high-quality solutions for multiple benchmark multi-objective test functions, such as *CEC2009*, *ZDT*, and *DTLZ* (*AbdelAziz et al., 2019*).

Recent advances in microarray technology allowed researchers to run thousands of experiments on multiple genes simultaneously, generating an enormous amount of data. These amounts of data comply to Big Data characteristics (*Demchenko et al., 2012*), thus raising the need to store, manage, and process genomic data with huge volumes (*Wong, 2016*). *Hadoop (2020)* and Spark *Guller (2015)* are Big Data technologies that provide both management and analysis for massive datasets. Spark is a programing framework that allows algorithms to run in parallel over distributed computing nodes (*Guller, 2015*). It uses the Resilient Distributed Dataset stored in memory to analyze distributed data, which ensures faster processing and avoids disk *I/O* burden encountered by Hadoop MapReduce (*Guller, 2015*). As a result, Spark can execute tasks that fit in memory 100 times faster than MapReduce. Even if data are larger than the cluster computing memory, Spark can run algorithms 10 times faster than MapReduce (*Guller,*

*2015*). These enhancements can enable Spark to minimize the running time of parallel algorithms to near real-time.

*MOWOATS* presented a good performance in solving multi-objective test functions (*AbdelAziz et al., 2019*), such as Zitzler-Deb Thiele (*ZDT*) (*Deb et al., 2002*), [−15]Deb-Thiele-Laumanns Zitzler (*DTLZ*) (*Deb et al., 2002*), and *CEC2009* test functions (*Zhang et al., 2009*). The method performance was evaluated according to the inverted generational distance metric (*Li & Yao, 2019*) and showed its ability to obtain solutions near Pareto front and evenly distributed over solution space. But a research question has appeared here, do we need to adapt to Big Data frameworks? Do we need to apply parallelization? We found that in order to run *MOWOATS* over massive datasets, it requires adaptation to Big Data frameworks. As Hadoop stores data in distributed nodes and Spark executes tasks in parallel, it is necessary to parallelize the components of *MOWOATS* to reap the benefits of these technologies. Parallelization ensures that *MOWOATS* can run faster than when it runs sequentially, which allows faster analysis for Big Data. In this article, *MOWOATS* is proposed to be applied in clustering microarray datasets based on three objective functions. These objective functions are developed to ensure finding the best set of clusters. Both intensification and diversification techniques are applied to ensure efficient coverage of the solution space of a *MOP*, while *PO* is used to identify non-dominated solutions. The main contributions of this paper can be summarized into:

- Redesign *MOWOATS* to parallelize its components and objective function to work over Hadoop and Spark frameworks.
- Test and analyze *MOWOATS* performance over small and medium-sized datasets according to both statistical methods (Silhouette index, Dunn index, Davies–Bouldin index) and a visual method (Eisen plot).
- Apply the parallelized version of *MOWOATS* over three real-life massive biological datasets to assess the performance of *MOWOATS* regarding two criteria: quality of obtained clusters and the scalability of the algorithm.

To assess the effectiveness of *MOWOATS* in analyzing microarray datasets, it has been applied over multiple real-life public small and medium-sized microarray datasets (*Maulik, Mukhopadhyay & Bandyopadhyay, 2021*). Quality of generated clusters has been measured using multiple validation methods, such as the Silhouette index (*Rousseeuw, 1987*), the Davies–Bouldin Index (*DBI*) (*Davies & Bouldin, 1979*), the Dunn Index (*DI*) (*Dunn, 1974*), and the Eisen plot (*Eisen et al., 1998*). These methods represent both statistical and graphical assessment metrics for generated clusters. Then, *MOWOATS* has been applied over massive microarray datasets to assess its performance according to the quality of generated clusters and the scalability of the algorithm. The microarray datasets used in the evaluations are all publicly available in the National Center for Biotechnology Information (*NCBI*) (*NCB, 2020*). The scalability of *MOWOATS* has been evaluated by running it on computing clusters with different number of nodes. Results showed that the running time was inversely proportional to the number of computing

**Table 1 An example of the first set of rows of the human fibroblasts serum microarray dataset** (*Maulik, Mukhopadhyay & Bandyopadhyay*).

| Genes | $t = 0$ h | $t = 15$ min | $t = 30$ min | $t = 1$ h | $t = 2$ h | ... | $t = 24$ h |
|---|---|---|---|---|---|---|---|
| *W95908* | 1.5962 | 0.534 | −1.8179 | −0.035017 | 1.8996 | ... | −0.22469 |
| *AA045003* | 0.095122 | 2.0874 | 0.26687 | 0.61037 | 0.85082 | ... | −0.76362 |
| *AA044434* | 0.84243 | 1.359 | 0.68745 | 0.84243 | 0.32584 | ... | −1.9471 |
| *W88572* | 1.1363 | 0.98137 | 1.1363 | 0.36156 | 0.30991 | ... | −0.67147 |
| *AA059077* | 0.23452 | 1.6489 | 1.6175 | 0.61169 | 0.54883 | ... | −1.777 |
| *AA035657* | 0.64514 | 1.2043 | 1.0179 | 0.64514 | 0.94334 | ... | −1.6286 |
| *AA180272* | 0.28406 | 1.3116 | 1.408 | 0.70151 | 0.28406 | ... | −1.5463 |

nodes. Also, the quality of generated clusters has not been affected with the size of the datasets, which reveals the efficiency of the modified *MOWOATS* algorithm, objective functions, and the programing code in reaping the benefits of Spark framework.

The article is organized as follows: the "Related Work" section gives a brief description of microarray technology and presents the recent work related to the proposed method. "Crossover Operator" describes the components of the algorithm, a pseudo-code of *MOWOATS*, a mathematical representation of the objective functions, and the selection criterion of the best solution. It also describes the modifications made to parallelize *MOWOATS* components and the objective functions. "Crossover Operator" reports the performance analysis of *MOWOATS* over small and mid-sized datasets according to the Silhouette index, *DBI*, *DI*, and the Eisen plot. Also, it reports the performance analysis of *MOWOATS* over real-life massive microarray datasets according to the quality of generated clusters and the algorithm scalability. "Crossover Operator" summarizes the main points of our work and the future work that we aim for.

## RELATED WORK

Microarray generates data that describe the expression profiles of samples being investigated during the experiment time (*Bolon-Canedo et al., 2014*). These data can be represented in the form of a matrix considering each sample as an instance and the sample status over different times as the features of each instance. Table 1 presents an example of the first set of rows of the Human Fibroblasts Serum microarray dataset (*Maulik, Mukhopadhyay & Bandyopadhyay, 2021*). Analyzing microarray data enables researchers to obtain valuable information, such as identifying correlated genes (*Chou et al., 2007*), evaluating the response of cells to a specific type of treatment (*Ban et al., 2011*), and identifying different types of cancer (*Saber & Elloumi, 2015*).

Clustering microarray data is not a trivial task. It operates over datasets with no prior information about labels of data objects (*Berry & Browne, 2006*). To improve the quality of clustering solutions, clustering methods developed different objective functions to improve the assessment of the homogeneity of data objects in each cluster and the separation among clusters (*Maulik, Mukhopadhyay & Bandyopadhyay, 2009*; *Bandyopadhyay, Mukhopadhyay & Maulik, 2007*; *Maulik, Bandyopadhyay & Mukhopadhyay, 2011*; *Acharya & Saha, 2018*; *Parraga-Alava, Dorn & Inostroza-Ponta,*

*2018*). This converted the clustering problem from a single objective problem to a multi-objective one (*Parraga-Alava, Dorn & Inostroza-Ponta, 2018*). An example of applying multiple objective functions to analyze microarray data was proposed by *Maulik, Mukhopadhyay & Bandyopadhyay (2009)*. The method proposed an improved clustering algorithm, based on two validity indices to assess the quality of the generated clusters. The best solutions were stored into a non-dominated set and a majority vote method was used to combine clustering information from all solutions stored in the non-dominated set. Genes were assigned to clusters with the highest membership degree. This reveals the importance of applying multi-objective validity indices to find best clusters.

To provide a faster coverage of the multi-objective solution space, researchers paid attention to the combination of *SI* methods with *PO*. Since (*Schaffer, 1984*) proposed combining *SI* methods with *PO* to solve *MOPs*, many *SI* methods have been developed for analyzing microarray data using this strategy (*Maulik, Bandyopadhyay & Mukhopadhyay, 2011*; *Mukhopadhyay, Maulik & Bandyopadhyay, 2015*; *Paul, Shill & Kundu, 2016*; *Mandal & Mukhopadhyay, 2017*; *Zareizadeh et al., 2018*). An example of these methods was a Multi-Objective Clustering algorithm Guided by a-Priori Biological Knowledge (*MOC-GaPBK*) for microarray data analysis, proposed by *Parraga-Alava, Dorn & Inostroza-Ponta (2018)*. The method focused on developing efficient intensification and diversification techniques to cover the solution space efficiently, and used *PO* to ensure an optimization of the whole objectives. *MOC-GaPBK* revealed the importance of employing intelligent methods to cover the solution space. *MOC-GaPBK* developed a gene ontology method to enhance the identification of similarity among genes. The method was tested over small and medium-sized datasets and presented its ability to obtain clustering solutions with good quality (*Parraga-Alava, Dorn & Inostroza-Ponta, 2018*). The method was not extended to be applied over large datasets.

Previous methods were tested over small and medium-sized microarray datasets (*Paul, Shill & Kundu, 2016*; *Mandal & Mukhopadhyay, 2017*; *Parraga-Alava, Dorn & Inostroza-Ponta, 2018*). Recent advances in microarray technology led to constructing microarray datasets for different illnesses and for different species (*NCB, 2020*). The volume of these microarray datasets is qualified as Big Data (*Demchenko et al., 2012*). This led researchers to pay attention to adapt *SI* methods to Big Data technologies, such as Hadoop MapReduce and Spark (*Guller, 2015*). These conceptual and computing frameworks can enable analytical algorithms to run in parallel over distributed computing nodes.

An attempt to apply clustering methods in analyzing large microarray datasets was proposed by *Hosseini & Kiani (2018)*. They proposed a Fuzzy Weighted Clustering algorithm based on Hadoop MapReduce (*FWCMR*). The *FWCMR* method was tested over multiple large microarray datasets, stored over distributed nodes and it was validated by applying multiple clustering validity indices to verify its effectiveness.

Due to the enhancements made in Spark, it has been a perfect replica of Hadoop MapReduce (*Guller, 2015*). This encouraged researchers to develop algorithms based on Spark instead of Hadoop MapReduce (*Guller, 2015*). *Hosseini & Kiani (2019)* proposed a

Distributed Density based Clustering approach that used Hesitant Fuzzy weighted correlation coefficient as a similarity measure (*DDHFC*). The method was adapted to run over Spark framework to avoid the delay of Hadoop MapReduce. The method running time was significantly less than the running time of the recently proposed MapReduce one. *Hosseini & Kiani (2019)* used a single objective to analyze microarray datasets. This approach can be improved by employing multiple objective functions to ensure obtaining high quality solutions in a smaller time.

## Background

This section presents a description of main components of *MOWOATS* algorithm, such as tabu search, *WOA*, intensification by crossover, and diversification by crossover procedures.

## Tabu search

*TS* is a single-point meta-heuristic method that has been first proposed by *Glover (1986)* as a global optimizer. *TS* employs memory elements to improve the coverage of solution space. *TS* stores best solutions in an Elite List *EL* to identify promising regions to be searched thoroughly. The main problem of solving *MOPs* using *TS* is the usage of a single solution to cover solution space, which is not applicable for real-life problems.

## Whale optimization algorithm

*WOA* is a *SI* method that mimics the behavior of the humpback whale to cover solution space, which has been proposed by *Mirjalili & Lewis (2016)*. *WOA* simulates the bubble-net technique to provide an effective exploitation of promising regions. For the exploration, *WOA* applies the communication mechanism employed by whales to cover vast areas in ocean. *WOA* incorporates the communication technique among swarm members to provide a better coverage of solution space (*Mirjalili & Lewis, 2016*). *WOA* has been applied over multiple optimization and engineering problems and showed itself as an efficient global optimizer. The complete pseudocode of *WOA* is presented in Algorithm 1.

## Crossover operator

Crossover is operator has been inspired by genetic algorithm (*Goldberg, 1989*) to improve the quality of obtained solutions and enhance the diversity of the population. Crossover is applied by combining the two solutions in a random way to generate a new solution. This operator has been incorporated in *MOWOATS* to improve both the intensification and diversification techniques. In intensification, *MOWOATS* applies crossover among swarm members and non-dominated solution in *EL* to ensure covering the promising regions in solution space. In the diversification, *MOWOATS* applies crossover among swarm members to generate new solutions that prevents *MOWOATS* from getting trapped in local optima (*AbdelAziz et al., 2019*).

---

**Algorithm 1  WOA algorithm.**

Initialize the whales population $X_i$ ($i$ = 1,2,…,$n$)

Calculate the fitness of each search agent

$X^*$ = the best search agent

**while** $t$ < maximum number of iterations **do**

  **for** each search agent **do**

    Update $a$, $A$, $C$, $l$, and $p$.

    **if** ($p$ < 0.5) **then**

      **if** ($|A|$ < 1) **then**

        Update the position of the current whale with a random position in the neighborhood of the best solution in the current swarm.

      **else if** ($|A| \geq 1$) **then**

        Select a random agent ($X_{rand}$) from current swarm.

        Update the position of the current whale with a random location in the neighborhood of the random agent $X_{rand}$.

      **end if**

    **else if** ($p \geq 0.5$) **then**

      Update the position of the current whale according to the bubble-net technique.

    **end if**

  **end for**

  Check if any search agent goes beyond the search space and amend it

  Calculate the fitness of each search agent

  Update $X^*$ if there is a better solution

  $t = t + 1$

**end while**

Return $X^*$

---

## THE PROPOSED METHOD

In this section, the main components of the proposed method are described. A pseudo-code of *MOWOATS* and a mathematical representation of the objective functions are given. *MOWOATS* generates multiple solutions for the problem, so the Silhouette index (*Rousseeuw, 1987*) is used to determine the best solution with the highest value.

### Objective functions

*MOWOATS* applies three clustering validity indices to assess the quality of the obtained centroids: *SSI* (*Wang et al., 2004*), Xie-Beni index (*Xie & Beni, 1991*), and overall cluster deviation (*Handl & Knowles, 2007*). These objective functions evaluate the homogeneity among data objects in each cluster and the separation among clusters. The *SSI* (*Wang et al., 2004*) is applied instead of the Silhouette index (*Rousseeuw, 1987*) because it can be computed in parallel in contrast with the Silhouette index. *SSI* (3)

**Table 2 Objective functions used to evaluate the quality of cluster centers.**

| Parameter | Definition | | Value |
|---|---|---|---|
| Xie-Beni index (*XB*) *Xie & Beni (1991)* measures the quotient between the total variance and the minimum separation of the elements in the clusters. | $XB = \dfrac{\sum\limits_{k=1}^{K}\sum\limits_{i=1}^{n} D^2(C_k, X_i)}{n \times min_{k \neq l} D^2(C_k, C_l)}$ | (1) | Minimization |
| Overall cluster deviation (*Dev*) *Handl & Knowles (2007)* is defined as the overall summed distances between genes and their corresponding cluster centroid. | $Dev = \sum\limits_{k=1}^{K}\sum\limits_{x_i \in C_k} D(C_k, X_i)$ | (2) | Minimization |
| Simple Silhouette Index (*SSI*) *Wang et al. (2004)* is used to evaluate the accuracy of assigning points to clusters. | $SSI = \dfrac{1}{n}\sum\limits_{i=1}^{n} ss(i); \quad ss(i) = 1 - \dfrac{a(i)'}{b(i)'}$ | (3) | Maximization |

computes the distance to the centroids of the clusters instead of the points of each cluster. *SS(i)* (3) represents the distance of each data point *i* to its cluster centroid where $a(i)'$ and $b(i)'$ (*Wang et al., 2004*) stands for the distance from the data *i* to its point to its nearest cluster centroid. The Xie-Beni index (1) evaluates the coherence among data objects in each cluster and separation among cluster centroids. Overall cluster deviation (2) measures the total distance among data objects and their centroids. Table 2 presents the objective functions and the mathematical definition of each function.

## Multi-objective whale optimization algorithm combined with tabu search

Due to the drawbacks of *WOA*, such as selecting the swarm leader (*Zhu et al., 2017*), getting trapped in local optima (*Wei et al., 2018*), and obtaining solutions that are not evenly distributed over solution space (*Wei et al., 2018*). *MOWOATS* has been proposed to combine the *WOA* with *TS* to obtain almost optimal solutions of *MOPs* (*AbdelAziz et al., 2019*). *MOWOATS* employed the memory elements of *TS* to store best solution in *EL*, which provided a better guidance to swarm members while covering the solution space. Also, *MOWOATS* applied crossover operator among swarm members to improve the diversity in population and with non-dominated solutions in *EL* to enhance the quality of swarm members.

*MOWOATS* (*AbdelAziz et al., 2019*) starts with setting the main parameters of the algorithm. First, a set of solutions are generated randomly from the expression profile dataset to represent the initial population. Objective values of swarm members are computed based on Eqs. (1)–(3) according to Algorithm 3. Then, the algorithm updates the *EL* with the best solutions from the initial population according to the dominance criterion of *PO*. The non-dominated solutions in *EL* are then used to guide the swarm members of *WOA*. This improves the ability of *MOWOATS* to avoid getting trapped in local optima as the leading whale is selected randomly from *EL*. Depending on the *p* parameter, *MOWOATS* determines whether to apply intensification or diversification techniques. *MOWOATS* increases the chances of diversification to ensure obtaining

solutions are uniformly distributed over the solution space. The solutions stored in *EL* are used to guide swarm members during both intensification and diversification phases, so that the algorithm can ultimately escape from local optima. In case that the number of iterations without improvement exceeds the parameter (Max_nonImprove), *MOWOATS* applies randomly either intensification using crossover or diversification using crossover procedures over the swarm members (*AbdelAziz et al., 2019*).

The crossover is conducted by selecting a random number of centers from a solution to be replaced by the same centers in another solution. The number of swarm members involved in this operation is randomly selected on the condition that this number does not exceed the half number of the swarm members. These procedures work to improve the diversity within solutions. At the end of the algorithm, the non-dominated solutions stored in *EL* are returned. The complete pseudocode of the MOWOATS algorithm is presented in Algorithm 2.

## Selecting the best solution

As explained above, the solutions in *EL* represent the best solutions that cannot be further enhanced, in the sense that improving a single objective in non-dominated solutions leads to minimizing the quality of other objectives. To select the best solution from *EL*, the Silhouette index (*S*) is used (*Rousseeuw, 1987*). The silhouette index can be computed for a point *i* as:

$$Sil(i) = \frac{b_i - a_i}{\max(a_i, b_i)} \tag{4}$$

where $a_i$ is the average distance among point *i* and points in the same cluster, $b_i$ represents the average distance among point *i* and points in other clusters. The total silhouette index is computed as the average for all the clusters' points, which is given by the following equation:

$$S(C) = \frac{1}{n} \sum_{i=1}^{n} s_i \tag{5}$$

The solution with the highest *S(C)* value is selected.

## Adapting *MOWOATS* with spark framework

Figure 1 presents a block diagram of the main components of *MOWOATS* algorithm after its adaptation to Hadoop and Spark frameworks. The original microarray data has been stored in the Hadoop computing cluster. Hadoop partitions the data randomly, each set of data instances are stored in a computing node. *MOWOATS* main components have redesigned to run in parallel instead of sequential execution. The main time consuming part in *MOWOATS* is the evaluation of centroids of each solution. This part has been programed to run in parallel over Spark framework (*Guller, 2015*). The centroids are taken by Spark to broadcast them to all computing nodes that store data instances. Then, computing nodes compute the distances among centroids and data instances in each node in parallel. The distances are then sent to the master to compute the Xie-Beni (1),

**Distributed Computing Cluster**

**Figure 1 A flow diagram of analyzing Microarray data using *MOWOATS* executed in parallel over the Spark computing cluster.**

overall deviation (2), and *SSI* (3) to assess the quality of each solution. This technique decreases the processing time as each node computes the distance for data instances stored in it. Also, it minimizes the traffic overhead over network as slave nodes return only the distances not the data instances themselves. In our implementation, the Spark dataframe has been used to hold the microarray dataset (*Guller, 2015*). This improved the scalability of the algorithm and ensured a better utilization of the data processing resources.

A pseudo code of the parallel objective function is given in 3. The function takes a solution $S$ as an argument. The function resides in the master node. The function starts by broadcasting the centroids of the solutions to all computing nodes that stores data instances in. Each computing nodes traverses the whole data instances stored in it to compute the distance between each data instance and the centroids. For each data instance, the distance value and the index of centroid with minimum distance value are stored. The computing nodes combines the distances and indices to send them back to the master node in (Key, Value) structure. The master node combines the whole distances and indices to compute the objective values according to Algorithm 3.

# NUMERICAL EXPERIMENTS

The proposed method was implemented in a virtual machine environment with host operating system Linux (UBUNTU 16.04 distribution), and programed in scala (*Odersky et al., 2004*) to be adaptable to the Spark framework (*Guller, 2015*). Datasets were stored based on *Hadoop (2020)*. The experiments were conducted over a computing cluster that consisted of 6 virtual machines: a single master and 5 slave nodes. Virtual machines

---

**Algorithm 2** *MOWOATS* algorithm.

**Initialization.**

Set *Np* to the number of whales, *K* number of clusters, empty *EL* holding non-dominated solutions, set *Max_nonImprove* to be maximum number of iterations without improvement, set *nob j* = 3 representing number of objectives, and initialize the parameters of the whale algorithm.

**for** *i* = 1,…,*Np* **do**

    Generate an initial solution randomly from the dataset.

    Compute the objective values of current solution in parallel applying Algorithm 3.

    Update *EL* according to *PO* dominance principle.

**end for**

**Main Loop.**

**for** *t* = 1,…,*MaxIt* **do**

    **for** *i* = 1,…,*Np* **do**

        Update *MOWOATS* parameters *a*, *A*, *C*, *l*, *p*.

        **if** ($p > 0.2$) **then**

            **if** ($|A| < 1$) **then**

                Update the position of the current whale $\overrightarrow{X_i(t)}$ with respect to a random solution selected from *EL*.

            **else if** ($|A| \geq 1$) **then**

                Update the position of current whale $\overrightarrow{X_i(t)}$ with respect to a random whale $\vec{X}_{rand}(t)$ from current swarm.

            **end if**

        **else if** ($p \leq 0.2$) **then**

                Update the position of current whale $\overrightarrow{X_i(t)}$ with respect to a random solution selected from *EL*.

        **end if**

        Compute objective values of current whale $\vec{X}_i(t)$ in parallel applying Equations (Xie-Beni (1), overall deviation (2), SSI (3)).

        Update *EL* according to *PO* dominance principle.

    **end for**

    **if** (number of iterations without improvement $\geq$ *Max_NonImprove*) **then**

        Set θ to a random value.

        **if** (θ < 0.5) **then**

            Apply crossover operator among swarm members and solutions in *EL*.

        **else**

            Apply crossover operator among swarm members.

        **end if**

    **end if**

**end for**

Return non-dominated solutions in *EL*

---

**Algorithm 3 Compute objective function *(S)*.**

Let solution $S$ be the input of the function, $C = \{C1, C2, \ldots, CK\}$ is the set of centroids of $S$, $K$ number of centroids, $X_i^f$ is a data instance $i$ with $f$ features.

Master: Broadcast centroids of solution to all computing nodes

**for** each computing node $j$ **do**

    **for** each data instance $X_i^f$ in computing node $j$ **do**

        Compute the distance $D$ among $X_i^f$ and all centroids.

        Set $I_{min}$, $D_{min}$ to be the index and distance value of centroid $C$ with least distance value to data instance $X_i^f$.

    **end for**

    Combine all $I_{min}$ and $D_{min}$ for each data instance.

    Return all $I_{min}$ and $D_{min}$ in (key, value) structure.

**end for**

Master: combine the $I_{min}$ and $D_{min}$ from all computing nodes and compute the objective values according to Equations (Xie-Beni (1), overall deviation (2), and *SSI* (3)).

Master: Return the objective values of solution $S$.

---

**Table 3 Main parameter values of MOWOATS.**

| Parameter | Definition | Value |
|---|---|---|
| *MaxIt* | Maximum number of iterations | 50 |
| *Np* | Population size | 15 |
| *max_EL* | Maximum number of solutions stored in elite list | 50 |
| *Max_NonImprove* | Maximum number of iterations without improvement | 2 |

were connected using a local network and the hardware configurations of the cluster were as follows:

- Master node (Name Node): 2 processors, RAM 8 GB, and Hard disk 30 GB.
- Slave node (Data Node): 1 processor, RAM 4 GB, and Hard disk 20 GB.

A single node was used to compare the running time between a single node and the computing cluster. The single node configurations were:

- Single node configurations: 8 processors, RAM 32 GB, and Hard disk 40 GB.

These virtual machines were hosted on the server of the faculty of telecommunication engineering, Vigo university (https://www.uvigo.gal/uvigo_en/Centros/vigo/lagoas_marcosende/enxeneiro_telecomunicacion.html), Spain.

## Parameters setting

Table 3 presents the values of the important parameters of *MOWOATS*. These parameters are used to adjust the performance of the algorithm. The *MaxIt* parameter represents the number of whales in each swarm, while *Max_NonImprove* parameter stands for the

**Table 4 Description of datasets.**

| Dataset | Genes | Features |
|---|---|---|
| Yeast sporulation | 474 | 7 |
| Yeast cell cycle | 384 | 17 |
| *Arabidopsis thaliana* | 138 | 8 |
| Human fibroblasts serum | 517 | 13 |
| Rat CNS | 112 | 9 |

maximum number of iterations without improvement, and *max_EL* parameter represents the maximum number of non-dominated solutions that can be stored in *EL*.

## Description of datasets

*MOWOATS* was applied on small and medium microarray datasets to evaluate its effectiveness in clustering gene expression profile datasets. Table 4 presents a description of the datasets: dataset name, genes, and features of each dataset (*Parraga-Alava, Dorn & Inostroza-Ponta, 2018*). These datasets are publicly available in *Maulik, Mukhopadhyay & Bandyopadhyay, 2021*. The selected elements are picked after preprocessing the datasets to select features with the highest variance to be investigated and ignoring the remaining features (*Parraga-Alava, Dorn & Inostroza-Ponta, 2018*; *Maulik, Mukhopadhyay & Bandyopadhyay, 2021*). Also, values in the datasets are normalized by applying different mathematical functions, such as $\log_2$ transformation ratios and root mean square functions (*Maulik, Mukhopadhyay & Bandyopadhyay, 2021*).

### *Preprocessing the biological datasets*

To avoid the exhausting pre-processing operations, the Spearman correlation coefficient $r_s$ was applied to assess the similarity among gene expression profiles (*Hauke & Kossowski, 2011*). The Spearman coefficient uses the rank of the expression values instead of the data values themselves, since relations among ranks are linear, which fulfills the condition to use correlation. The Spearman coefficient was programmed to be computed in parallel over the distributed computing nodes. For any two gene expression profiles $G_x$ and $G_y$, the Spearman coefficient can be computed as:

$$d_i = Rank(G_{xi}) - Rank(G_{yi}) \tag{6}$$

$$r_s = 1 - \frac{6 \sum\limits_{i} d_i^2}{n(n^2 - 1)} \tag{7}$$

where $d_i$ represents the difference in the ranks and $n$ is the number of values.

Datasets were first pre-processed to get the rank matrices of the original datasets. *MOWOATS* was then executed over both the original and the rank matrices. Original matrices were used to obtain centers while rank matrices were used to measure the similarity among centroids and points. The rank was based on the minimum order, which assigned the same minimum rank to data with the same values. This operation

**Table 5 Characteristics of biological datasets obtained from NCBI.**

| Dataset | Samples | Features | Classes | GEO | |
|---------|---------|----------|---------|-----|---|
| EPI | 29 | 22283 | 2 | (16, 13) | (GSE15568) |
| SPC3 | 73 | 14010 | 3 | (13, 24, 36) | (GSE7458, GSE14869, GSE5147) |
| SPC4 | 79 | 57381 | 4 | (13, 24, 36, 6) | (GSE7458, GSE14869, GSE5147, GSE14275) |

enhanced the similarity evaluation among gene expression profiles, specially for datasets generated from different species as it will be seen in the next subsections.

### Massive biological datasets

To assess the performance of *MOWOAST* in analyzing huge microarray datasets, it was applied over publicly available datasets located in the National Center of Biotechnology Information (NCBI) (*NCB, 2020*). These datasets represent the expression profiles of a set of genes when exposed to different experiments in microarray. So, three large biological datasets were used to evaluate the performance of *MOWOATS*. The first dataset accession number is GSE15568. The dataset gave a study of gene expression profiles of rectal epithelia of cystic fibrosis for 29 patients, each sample had 22283 features. The dataset had 2 classes (patients carrying the Cystic Fibrosis-specific D508 mutated CFTR-allele (CFTR-D508) compared with gene expression profiles of normal ones). For simplicity, the dataset will be named *EPI* in the rest of the paper.

To provide a better assessment of *MOWOATS*, it was applied over datasets with cross-species. Two datasets were used for this evaluation. The first dataset was composed of three different species that were discussed by *Kristiansson et al. (2013)*. The first cross-species dataset consisted of 3 species: (1) *Homo sapiens* dataset with accession number GSE7458, (2) *Mus musculus* dataset with accession number GSE14869, (3) *Drosophila melanogaster* dataset with accession number GSE5147 (*Kristiansson et al., 2013*). For simplicity, the dataset will be named *SPC3* in the rest of the article. The second cross-species dataset consisted of the previous three datasets added to them *Oryza sativa* dataset with accession number GSE14275. For simplicity, the dataset will be named *SPC4* in the rest of the article. A complete description of the three datasets is given in Table 5. It shows a description of each dataset: number of samples, number of experimental conditions that samples were exposed to, number of classes in each dataset, and accession numbers of datasets in *NCBI*.

### Evaluation criteria

To provide a solid assessment of the obtained clusters, several cluster evaluation metrics have been applied over the generated ones. These metrics include: Silhouette index (*Rousseeuw, 1987*), DBI (*Davies & Bouldin, 1979*), DI (*Dunn, 1974*), and *F*-measure (*Dalli, 2003*). These metrics aim to evaluate the homogeneity of samples/genes in the same cluster and the separation of samples/genes in different clusters. The Silhouette index is an important index for measuring the quality of a clustering partition. It measures the cohesion and separation among clusters over both point and cluster level. It assesses the similarity of each point for both points in the same cluster and points in other clusters.

The silhouette index can be computed according to Eqs. (4) and (5). Silhouette is a maximization index, the bigger the silhouette value, the better the clustering solution (*Bouyera & Hatamlou, 2018*).

Moreover, the *DBI* has been used to measure the compactness of generated clusters and how well these clusters are separated. *DBI* can be computed by obtaining the ratio of within-cluster distance and between-cluster separation according to:

$$DBI = \frac{1}{m} \sum_{i=1}^{m} \max_{i \neq j} \left( \frac{S_i + S_j}{M_{i,j}} \right) \quad 1 \leq i, j \leq m, i \neq j \tag{8}$$

Here, $S_k$ represents within-cluster distance in cluster $k$, and $M_{i,j}$ stands for between-cluster separation.

*DBI* is a minimization function, the smaller the value of *DBI* the better the quality of obtained solution (*Davies & Bouldin, 1979*).

Additionally, the *DI* has been used to evaluate the compactness of each cluster and separation among clusters. *DI* can be computed as:

$$DI = \frac{\min \delta(C_i, C_j)}{\max \Delta_k}, \quad 1 \leq i, j, k \leq m, i \neq j \tag{9}$$

where $m$ represents the number of clusters, $\delta(C_i, C_j)$ stands for inter-cluster distance between clusters $C_i$ and $C_j$, and $k$ is the maximum distance between two points in the same cluster. Note that *DI* is a maximization function, the bigger the value of *DI* the better quality of obtained solution.

Finally, the *F*-measure (*Dalli, 2003*) criterion has been used to provide an outlier assessment of generated clusters compared to original classes. The higher the value of *F*-measure, the better the quality of the generated clusters. *F*-measure depends on combining both precision and recall used in information retrieval. *F*-measure represents generated clusters as $C_j$ for $j = 1,...,K$. Each cluster $j$ consists of $n_j$ instances. The number of instances that belong to class $i$ are represented by $n_i$. The number of instances in class $i$ and belongs to the cluster $j$ are represented by $n_{ij}$. Precision $p(i, j)$ and recall $r(i, j)$ are defined respectively by

$$r(i,j) = \frac{n_{ij}}{n_i} \quad \text{and} \quad p(i,j) = \frac{n_{ij}}{n_j}, \quad \forall i, j \tag{10}$$

The *F*-measure value $F(i, j)$ is

$$F(i,j) = \frac{2p(i,j)r(i,j)}{r(i,j) + p(i,j)} \tag{11}$$

At the end, the *F*-measure value of the whole dataset that consists of $n$ data instances divided into $K$ clusters is computed as:

$$F = \sum_{i=1}^{K} \frac{n_i}{n} \max_j F(i,j) \tag{12}$$

**Table 6 Mean values of Silhouette index over 20 runs of different algorithms.** The best results are presented in bold.

| Algorithm | Arabidopsis | Cell cycle | Sporulation | Serum |
|---|---|---|---|---|
| *MOWOATS* | **0.6** ($k = 6$) | **0.64** ($k = 6$) | **0.81** ($k = 6$) | **0.69** ($k = 6$) |
| *MOC-GaPBK* | 0.49 | 0.63 | 0.80 | 0.58 |
| *Semi-FeaClustMOO* | 0.46 | 0.50 | 0.70 | 0.44 |
| *MO fuzzy* | 0.41 | 0.43 | 0.59 | 0.40 |
| *MOGA* | 0.40 | 0.42 | 0.58 | 0.38 |
| *SOM* | 0.23 | 0.38 | 0.58 | 0.34 |
| *Avg. link.* | 0.32 | 0.44 | 0.50 | 0.36 |

## RESULTS AND DISCUSSION

*MOWOATS* was compared with *MOC-GaPBK* algorithm that used new modified intensification and diversification strategies to provide a good coverage for the solution space (*Parraga-Alava, Dorn & Inostroza-Ponta, 2018*). *MOC-GaPBK* used two functions to measure the similarity among gene expression profiles: Pearson correlation coefficient (*Hauke & Kossowski, 2011*) and biological knowledge (*Wang et al., 2007*). The Wang functional similarity was applied to measure the biological similarity between two genes based on ontology terms (*Wang et al., 2007*). To evaluate the quality of generated clusters, three objective functions were used: Xie-Beni index, overall cluster deviation, and cluster separation (*Parraga-Alava, Dorn & Inostroza-Ponta, 2018*).

A comparison among *MOWOATS*, *MOC-GaPBK*, *Semi-FeaClustMOO*, *MO-fuzzy*, *MOGA*, *SOM*, and average linkage clustering techniques (*Parraga-Alava, Dorn & Inostroza-Ponta, 2018*) is presented in Table 6. These results present the mean Silhouette index values for each method averaged for 20 different runs. Results were obtained from running the algorithms over real-life gene expression datasets: *Arabidopsis thaliana*, Yeast Cell Cycle, Yeast Sporulation, and Human Fibroblasts Serum. The class labels of data objects in the datasets were not known in prior, so the algorithms were executed for different number of clusters $K \in \{4, 5, 6\}$.

Results in Table 6 show that *MOWOATS* achieved the best Silhouette index values over all other methods for all datasets. *MOC-GaPBK* reached a Silhouette index value that is close to the one found by *MOWOATS* for the sporulation dataset and in general, it was the second best algorithm over the whole datasets. This emphasizes the importance of the modified search methods to cover effectively the solution space. The good results achieved by *MOWOATS* returns to its improved search capabilities that drove it to cover the solution space effectively. Also, the objective functions enabled *MOWOATS* to evaluate precisely the quality of the solutions during the search process. Although *MOWOATS* did not use the biological knowledge as a similarity measure, it was able find solutions better than those found by *MOC-GaPBK*, highlighting the efficiency of the implemented search techniques.

Expression profiles represent the reaction of genes to different experiments. Microarray collects the reactions of samples to the predefined experiments in pre-determined time intervals. Similar samples react in the same way, so different/malignant samples tend to

react in a different manner than normal ones. *Eisen et al. (1998)* plot based on a heat map has been applied to depict the homogeneity expression profiles that have been generated of *MOWOATS*. Figure 2 presents a graphical representation of the best two clustering solutions with the highest Silhouette index values. Each row depicts the reaction of a specified gene to the same experiment over different time intervals. The more similar colors grouped, the better the quality of the generated clusters. As shown in the figure, the rows in each cluster are similar to each other over the different time intervals.

Figure 2 shows the homogeneity of samples in each cluster, which clarifies the effectiveness of *MOWOATS* in analyzing microarray datasets. Also, this ensures the correctness of using multiple objective functions to analyze complicated datasets like microarray data.

*FWCMR* was a recent attempt to analyze microarray datasets by developing a new similarity measure, which was based on using the Spearman correlation coefficient (*Hauke & Kossowski, 2011*). The method applied the density-based clustering to analyze the microarray data (*Hosseini & Kiani, 2018*). *FWCMR* was evaluated over different microarray datasets and it was able to obtain good clustering solutions. *Hosseini & Kiani (2018)* employed different clustering validity indices to assess the quality of the generated clusters. They used the *DI*, *DBI*, and Silhouette index as clustering validity indices to provide a more trustful evaluation of the obtained clustering solutions.

Tables 7–9 present comparisons among *MOWOATS*, *FWCMR*, Rough-Fuzzy Clustering (*RFC*) (*Maji & Paul, 2013*), Modelling Based Clustering (*MBC*) (*Blomstedt et al., 2016*), Multi-objective Symmetry Based Clustering (*MSBC*) (*Saha et al., 2013*), and Hessian Regularization Based on Symmetric Clustering (*HRSC*) (*Ma et al., 2016*) according to three clustering validity indices *DI*, *DBI*, and *S* (*Hosseini & Kiani, 2018*), averaged for 16 different runs. The values presented in the table have been obtained from the original articles.

Results in Tables 7–9 presented the superiority of *MOWOATS* over other methods for all datasets. *MOWOATS* was able to obtain clusters with the highest *DI* and *SI* values and with the least values of *DBI*. This proves the homogeneity of the obtained clusters and corroborates the effectiveness of applying multiple objective functions to analyze microarray datasets.

Additionally, Table 10 presents the validity indices: *F*-measure and Silhouette index values for the three biological datasets. The table shows the average and standard deviation values for the evaluation criteria, averaged for 20 different runs. For the cross-species datasets, *MOWOATS* achieves very good results and generates clusters that are equal to or very similar to the original classes. This clearly shows the effectiveness of *MOWOATS* in analyzing microarray datasets with huge volumes. For the *EPI* dataset, *MOWOATS* also achieved reasonable results. This returns to the success of the objective functions in evaluating clustering solutions and the effectiveness of the Spearman coefficient in assessing the similarity among samples.

To further verify the effectiveness of *MOWOATS* in analyzing huge microarray datasets, it was compared to clustering methods that were based on MapReduce and Spark frameworks, such as a MapReduce based *K*-means method (*MRK*) proposed by

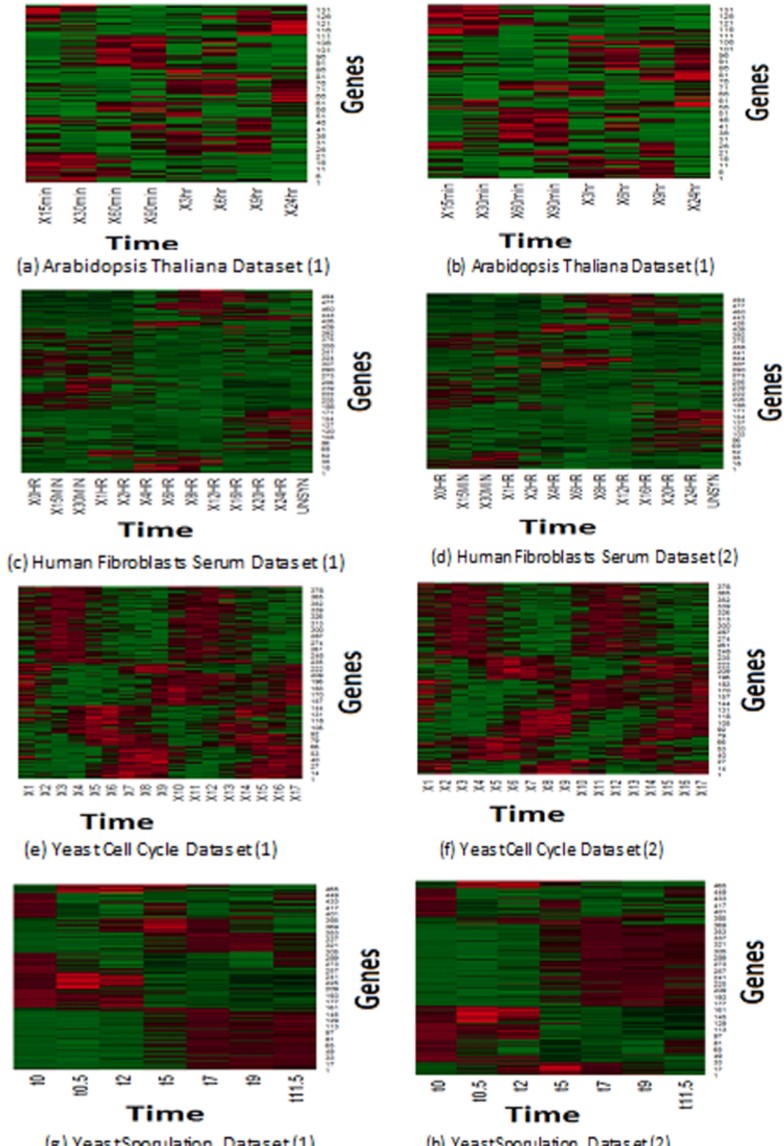

**Figure 2** (A–H) Depiction of the best two solutions obtained by *MOWOATS* for four real-life datasets according to the Eisen plot.

*Shahrivari & Jalili (2016)* and MapReduce based Bee colony clustering method (*MRB*) proposed by *Banharnsakun (2017)*. Articles that represent clustering methods based on Spark were: *K*-M algorithm presented in Spark Machine learning Library (*MLK*) proposed by *Gopalani & Arora (2015)*, Spark DBscan algorithm proposed (*SDB*) by *Luo et al. (2016)*, and *DDHFC* proposed by *Hosseini & Kiani (2019)*. All these methods combined *SI* methods with Big Data frameworks. Table 11 presents the comparison among *MOWOATS* and the methods enumerated before according to S, *DI*, and *DBI* evaluation criteria when applied over the *SPC4* dataset as reported by *Hosseini & Kiani (2019)*.

As seen in Table 11, *MOWOATS* dominates the other methods according to the three evaluation indices. *DDHFC* achieves the second-best results for all validity indices.

**Table 7 Comparison among *MOWOATS* and different clustering methods according to clustering validity indices (*DI, DBI, S*) achieved from *Arabidopsis thaliana* dataset.** The best results are presented in bold.

| Method | DI | DBI | S |
|---|---|---|---|
| RFC | 0.579 | 0.945 | 0.359 |
| MBC | 0.586 | 0.937 | 0.368 |
| MSBC | 0.610 | 0.918 | 0.399 |
| HRSC | 0.592 | 0.926 | 0.395 |
| FWCMR | 0.604 | 0.904 | 0.412 |
| MOWOATS | **0.678** | **0.767** | **0.6** |

**Table 8 Comparison among *MOWOATS* and different clustering methods according to clustering validity indices (*DI, DBI, S*) achieved from human fibroblasts serum dataset.** The best results are presented in bold.

| Method | DI | DBI | S |
|---|---|---|---|
| RFC | 0.450 | 0.903 | 0.364 |
| MBC | 0.432 | 0.911 | 0.359 |
| MSBC | 0.464 | 0.885 | 0.397 |
| HRSC | 0.458 | 0.897 | 0.381 |
| FWCMR | 0.483 | 0.869 | 0.452 |
| MOWOATS | **0.635** | **0.833** | **0.699** |

**Table 9 Comparison among *MOWOATS* and different clustering methods according to clustering validity indices (*DI, DBI, S*) achieved from Rat CNS dataset.** The best results are presented in bold.

| Method | DI | DBI | S |
|---|---|---|---|
| RFC | 0.291 | 0.814 | 0.439 |
| MBC | 0.288 | 0.822 | 0.425 |
| MSBC | 0.302 | 0.803 | 0.477 |
| HRSC | 0.313 | 0.794 | 0.482 |
| FWCMR | 0.320 | 0.782 | 0.496 |
| MOWOATS | **0.694** | **0.658** | **0.711** |

**Table 10 *F*-measure and Silhouette Index values for best 2 clustering solutions generated from *MOWOATS*.**

| Dataset | F-measure | Std. Dev. | S | Std. Dev. |
|---|---|---|---|---|
| EPi (2 classes) | 0.7241 | 0.00102 | 0.5944 | 0.00367 |
| | 0.6904 | 0.00453 | 0.5574 | 0.00138 |
| SPC3 (3 species) | 1.0 | 0.00517 | 0.9507 | 0.00233 |
| | 0.972 | 0.00237 | 0.9455 | 0.0095 |
| SPC4 (4 species) | 0.9627 | 0.00637 | 0.9416 | 0.00836 |
| | 0.8909 | 0.00233 | 0.9393 | 0.00639 |

Table 11 **A comparison among *MOWOATS, DDHFC,* and recently proposed methods regarding quality of generated clusters over *SPC4* dataset.** The best results are presented in bold.

| Method | S | DI | DBI |
|---|---|---|---|
| *MLK* | 0.56 | 0.49 | 0.55 |
| *SDB* | 0.63 | 0.54 | 0.43 |
| *MRK* | 0.57 | 0.50 | 0.56 |
| *MRB* | 0.61 | 0.52 | 0.48 |
| *DDHFC* | 0.82 | 0.67 | 0.32 |
| *MOWOATS* | **0.94** | **0.78** | **0.25** |

Table 12 **The execution times of the MOWOATS algorithm in minutes over different number of computing nodes for the three massive datasets (*SPC3, EPI, SPC4*).**

| No. of Nodes | *SPC3* | *EPI* | *SPC4* |
|---|---|---|---|
| 6 | 19.607 | 28.75 | 457 |
| 4 | 25.909 | 33.65 | 551 |
| 2 | 34.28 | 37.82 | 623 |
| 1 | 42 | 44.15 | 680 |

The performance gain with *MOWOATS* confirms the correctness of using multiple objective functions in analyzing microarray datasets. This also presents the importance of the memory elements used in *MOWOATS*. Also, results in Tables 10 and 11 show that the quality of clusters obtained by *MOWOATS* is very high even for large datasets. This ensures the stability of *MOWOATS* in obtaining high quality solutions regardless of the size of the dataset.

Moreover, to assess the scalability of *MOWOATS* when applied over computing clusters, it was tested over a computing cluster that consisted of one master node and 5 slave nodes. The number of nodes was varied to assess the scalability of the method over (1, 2, 4, 6) nodes. Table 12 presents the execution times of the algorithm over the three massive datasets according to different number of computing nodes, averaged for 20 different runs. Figures 3 and 4 present the normalized running time of *MOWOATS* over different number of computing nodes for the three datasets. The figure shows the near-linear decrease in running time as the number of nodes increases. This proves the capability of *MOWOATS* to minimize the running time needed to analyze huge datasets. The figure also presents the high-quality of the programing code that could minimize the number of sequential loops to take the advantages of the Spark framework.

From previous discussion, *MOWOATS* presented itself as an effective tool to analyze huge microarray datasets. It was able to obtain clusters that were very similar to the classes of the original datasets. This shows the effectiveness of the objective functions used to evaluate clustering solutions. Also, *MOWOATS* running time was inversely proportional to the number of computing nodes, which shows its high scalability.

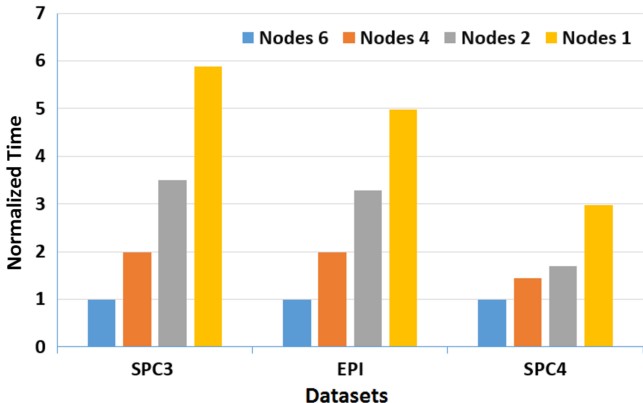

**Figure 3** A depiction of normalized running time of *MOWOATS* for SPC3, EPI, and SPC4 datasets over a different number of computing nodes.

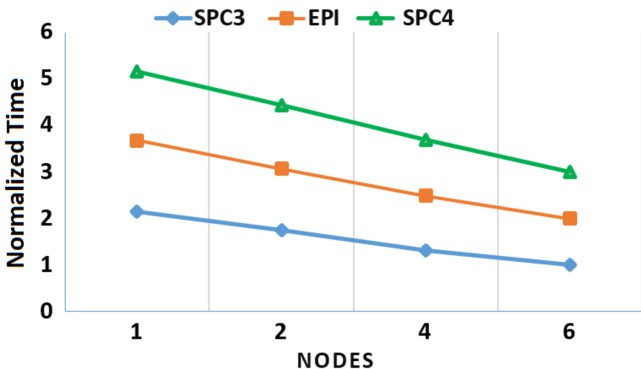

**Figure 4** A depiction of the speedup gain of *MOWOATS* when the number of computing nodes is increased for the SPC3, EPI, and SPC4 datasets.

## CONCLUSION AND FUTURE WORK

Microarray has been a revolutionary tool that generates vast volumes of data that describe the expression profiles of genes being investigated. It exposes thousands of genes to different conditions in a single experiment. The sheer volume of generated data can be qualified as Big Data. Analyzing genomic datasets with huge volumes can allow researchers to obtain valuable information, such as identifying correlated genes and predicting the response of patients to certain medications on the genomic level. In this paper, a hybrid Multi-Objective (*MO*) algorithm that combined Whale Optimization Algorithm with Tabu Search (*MOWOATS*) was proposed for analyzing huge microarray datasets. *MOWOATS* used three objective functions to measure the quality of obtained solutions: Simple Silhouette index, Xie-Beni index, and overall distribution of data objects. These objective functions ensured that the obtained clusters have the highest coherence among data objects in each cluster and maximum separation among clusters. *MOWOATS* components were modified to run in parallel over Big Data frameworks. To assess the efficiency of *MOWOATS*, it was applied over public real-life small and medium-sized microarray datasets. The obtained clusters were evaluated both statistically and

graphically. It achieved good results for statistical evaluations, which expose the coherence of generated clusters. Also, the graphical evaluation presented the unification of data objects in each cluster. To assess its performance in analyzing huge microarray datasets, *MOWOATS* was applied over three large public real-life microarray datasets. The performance of *MOWOATS* was assessed according to the quality of the generated clusters and its scalability. Generated clusters presented a great coherence and were very similar to classes of original datasets. Also, the running time was inversely proportional to the number of computing nodes, which ensured high scalability. This presents *MOWOATS* as an effective microarray data analysis tool that can be used in real-life applications. Despite the size of the datasets, the algorithm could minimize radically the running time by increasing the number of computing nodes. For future work, apply *MOWOATS* over more computing nodes using cloud computing to minimize the execution time and to prepare *MOWOATS* for bigger datasets. Also, we aim to add gene ontology methods to provide a better evaluation of generated solutions, which prepares *MOWOATS* to be applied in advanced analysis of real-life microarray datasets.

## ACKNOWLEDGEMENTS
The authors thank the group involved in the joint project "*KA107*" between Vigo University and Beni-Suef university.

### Funding
The authors received no funding for this work.

### Competing Interests
The authors declare that they have no competing interests.

### Author Contributions
- Amr Mohamed AbdelAziz conceived and designed the experiments, performed the experiments, analyzed the data, performed the computation work, prepared figures and/or tables, authored or reviewed drafts of the paper, and approved the final draft.
- Taysir Soliman performed the experiments, analyzed the data, authored or reviewed drafts of the paper, and approved the final draft.
- Kareem Kamal A. Ghany performed the experiments, prepared figures and/or tables, authored or reviewed drafts of the paper, and approved the final draft.
- Adel Sewisy performed the computation work, prepared figures and/or tables, and approved the final draft.

### Data Availability
Yeast Sporulation, Yeast Cell Cycle, *Arabidopsis thaliana*, Human fibroblasts serum, Rat CNS data are available at: http://anirbanmukhopadhyay.50webs.com/data.html.

Additional data is available at NCBI: GSE7458, GSE15568, GSE7458, GSE14869, GSE5147, GSE14275.

## Supplemental Information

Supplemental information for this article can be found online at http://dx.doi.org/10.7717/peerj-cs.416#supplemental-information.

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
