# Peer review of "A hybrid multi-objective whale optimization algorithm for analyzing microarray data based on Apache Spark"

_PeerJ Computer Science, doi:10.7717/peerj-cs.416_

## Round 0.1 · original submission · Major Revisions

Two of the reviewers ask for significant clarification and changes in the manuscript. Please address their comments one by one.

Reviewer 1 ·

Basic reporting

The paper titled “A Hybrid Multi-Objective Whale Optimization Algorithm for Analyzing 2Microarray Data based on Apache Spark” is well structured and written. But there are some minor corrections to be carried out for publications:
1. Clear and unambiguous professional used throughout.
2. Professional article structure figures raw data shared.
3. Some additional information regarding background context need to be provided.

Experimental design

No comments

Validity of the findings

No comments

Additional comments

In the conclusion and future work, the author has not provided some future work relevant of the research carried out. It would be better if the author also provides some limitations of the proposed work which can be carried out as future work.

Reviewer 2 ·

Basic reporting

The structure of the paper should be improved to clearly differentiate previous work from the main contribution. All components of the MOWOATS algorithm should be described (solution initialization, crossover operator, selection mechanism…) in a background section prior to presenting the changes made to adapt MOWOATS to the multi-objective gene clustering problem (solution representation, objective functions and distributed implementation).

I strongly suggest including a methodology section to explain the main goal of each experiment, the characteristics of the datasets, the evaluation metrics, and the comparison methods. In particular, the authors should justify the choice of algorithms for comparison, including a short description of them, and full details of their parameter set up. Now this information is missing or blurred among the experimental sections.

Execution time should be provided in absolute terms too. Normalized values give useful information about relative improvements but hide the time unit which is also important. It is not the same to reduce from 6 days to 1 day, or from 6 seconds to 1 second. That will help to understand if the effort and cost in infrastructure is paid off. Speed-up rates could be computed as well to enrich the analysis of results.

The adaptation of MOWOATS to the Spark framework should be extended with more details, e.g. some pseudocode. This is one of the main contributions but explained in only 10 lines (207-217).

Experimental design

The manuscript does not include explicit research questions, what makes it difficult to understand the purpose of the experiments. Research questions usually help to guide the discussion of the main findings.

The algorithm is not completely new, and the problem domain has been addressed before too, so the focus of the paper should be put on the experiments. I highlight the number of algorithms selected for comparison and the use of public datasets of different complexities. However, some aspects of the experimental design are not clear:

- Explain why the evaluation metrics for small/medium datasets are different from large dataset.
- Try to apply all methods to all datasets unless it is not technically possible for any reason. In such a case, mention it explicitly.
- If execution time is relevant, and it is in this work, comparison methods should be executed under the same platform as well. Current comparison is only provided in terms of clustering performance.
- It is not clear whether the authors execute the comparison methods or their results are simply gathered from the original papers.
- When and to which datasets preprocessing is applied. Is that common for all methods or required by MOWOATS only?
- Minimum number of repetitions recommended for metaheuristics is 30 or more. Less runs could be valid in costly experiments but decrease technical soundness and should be discussed.

Validity of the findings

Clustering performance should be statistically assessed to confirm the significance of the results, and it is especially important when algorithms are randomized. In fact, it is not clear whether authors show best results (one run) or the average of 20 runs. In the second case, standard deviation should be included too.

Conclusions about execution time are somehow expected, and have less scientific value, because they are not compared to any baseline. The datasets seem not be so huge as initially stated, so the authors should clarify how much time a standard algorithm in one CPU could need to cluster them. It is true that the number of features is considerably high, but very few samples are included in each dataset, even in the largest ones. A reader needs more evidence to believe the problem is truly computationally expensive.

From my point of view, results should be analyzed from a trade-off perspective, as improvement in performance might not be so important if execution time and the infrastructure needed to run the method is prohibitive.

Additional comments

The paper addresses the problem of clustering gene expression profile by means of a whale optimization algorithm enhanced with tabu search and adapted to a multi-objective perspective. The main contribution is the implementation of the algorithm, which was previously proposed by the authors, in Spark. The technical contribution is good, but the paper lacks a more scientific contribution. Besides, according to the authors, other proposals of bio-inspired clustering implemented in Spark or other platforms already exist, so even the technical contribution might not be so novel. A series of experiments with different real-world and publicly available datasets is presented, and results are compared with similar algorithms. The authors provide source code, what is valuable for reproducibility purposes.

Some other comments:

1) The “Big Data” problem might not be so realistic considering the datasets finally used. Are these datasets the only available? For instance, the cited work by Hosseini and Kiani [1] mentions a synthetic dataset with up to 1 million genes.

2) Last paragraph in the introduction is not readable as numbers of sections do not appear.

3) PeerJ Computer Science is not a specialized journal for metaheuristics, so a short introduction to whale optimization could be useful for non-familiar readers.

4) Details of the tabu search (TS) hybridization are not provided but using this kind of search in a multi-objective algorithm is not trivial [2,3]. Even if it is part of previous work, the authors should briefly explain when TS is applied, how solutions from the swarm are chosen for local improvement, and the acceptance criterion when multiple objectives exist. I did not see nothing specific of TS in Algorithm 1 and its explanation.

5) Multi-objective algorithms are compared in terms of quality indicators that measure convergence and diversity of the Pareto front [4]. However, here only one solution is chosen as final solution for comparison, so the need of a multi-objective approach is not evident. The authors could have optimized a main objective, with secondary objectives, or aggregate values somehow if they do not want to provide the user with a set of solutions for his/her choice.

6) The algorithm looks for diversification by means of a crossover operator, which is not described in detail. For genetic algorithms, it is known that crossover promotes convergence, whereas mutation is better for diversification. To the best of my knowledge, standard whale optimization does not have these kinds of operators, as it works similar to particle swarm optimization (PSO), i.e. by updating swarm member positions. PSO has something similar to mutation to “disturb” the swarm and promote diversity, so I guess whale optimization might work under the same assumptions.

7) Figure 2 is not easy to interpret for reader without background on gene expression profiles. They should be explained in more detail.

8) Sentence in lines 352-354 claims about the quality of code, but this is something that cannot be inferred from the figure.

[1] B. Hosseini, K. Kiani. “FWCMR: A scalable and robust fuzzy weighted clustering based on MapReduce with application to microarray gene expression”. Expert Systems with Applications, vol. 91, pp. 198-210. 2018.
[2] N. Krasnogor, J. Smith. “A Tutorial for Competent Memetic Algorithms: Model, Taxonomy, and Design Issues”. IEEE Transactions on Evolutionary Computation, vol. 9, no. 5, pp. 474–488, 2005.
[3] A. Jaszkiewicz, H. Ishibuchi, Q. Zhang. “Handbook of Memetic Algorithms”. Springer Berlin Heidelberg, 2012, ch. Multiobjective Memetic Algorithms, pp. 201–217.
[4] M. Li, X. Yao. “Quality evaluation of solution sets in multiobjective optimisation: A survey”. ACM Computing Surveys, article 26, March 2019.

Reviewer 3 ·

Basic reporting

This paper adapts an existing multi objective whale optimisation / tabu search algorithm for the Apache Spark framework, and applies it to microarray data. The paper is generally structured ok, and the experimental results look promising. The biggest problem is the lack of detail on the algorithm adaptation for Spark, one of the main contributions of the paper.

Literature review / background
- The literature review should include more background on Spark and why Spark is often chosen over Hadoop these days, for people unfamiliar with big data technologies.

Proposed method:
- description of the MOWOATS algorithm is ok, inclusion of pseudocode is good. Could motivate clearer why tabu search and whale optimisation are good for multi-objective problems.

Minor problems:
- Cross references within the manuscript need checking; e.g. in the introduction lines 106-114, the Section numbers are not showing.
- The references tend to blend into the text, as there are no brackets around the author names – this might be a document template problem? But it should be fixed, as it makes the text very difficult to read.
- Good that the code is provided as a supplemental file, but it should be better documented

Experimental design

The introduction clearly states the three main contributions of the paper. The second and third contributions (lines 87-92) are covered by the experimental section.

However, the main body of the paper contains very little on the first contribution - adapting MOWOATS for the Spark framework. Lines 207-217 are not sufficient, and this subsection needs to be significantly expanded. At a minimum, I would suggest including:
- how the data is divided amongst the computing nodes
- description of how the algorithm is adapted to run in Spark, including what computation is done on the individual nodes, and how are the results aggregated at the end.
- Spark 'pseudocode' showing the flow of the Spark implementation of the algorithm and Spark operations used.

The experimental results section could be improved by being organised better, such as:
- the entire experimental framework could be detailed at the beginning, including description of datasets, any preprocessing done, algorithms compared against etc., then below that give the results
- It is unclear what the section 'Preprocessing the biological datasets' (line 293) is describing, or why it is needed

Validity of the findings

- All datasets are publically available
- the results generally look promising - they show the advantage of MOWOATS for microarray data over other algorithms, and that the algorithm can be effectively implemented in Spark.
- for the results on the larger datasets, I don't think any of the comparison algorithms are designed for microarray data specifically. So, it would be interesting if you could also implement some of the methods from the smaller dataset experiments in Spark, and apply them to the larger datasets. This would give a better sense of the performance of the algorithm on large datasets, in terms of the quality of the generated clusters.

---

## Round 0.2 · Minor Revisions

Please address the minor revision comments of the reviewer

Reviewer 2 ·

Basic reporting

The paper now includes a background section as requested. The section introduces the applied techniques (tabu search, whale optimization), but I cannot find any paragraph explaining how tabu search is integrated in MOWOATS. The only element later mentioned is the elitist list, but this is a general mechanism in metaheuristics to keep best individuals, not a particular component of tabu search. I still fail to see how the authors apply multi-objective tabu search to solutions generated via whale optimization. I refer to the definition of neighborhood for this problem, which solutions are compared (best/random choice) and how (Pareto-based, objective aggregation, one objective only…) and which problem characteristics are considered to build and update the tabu list (solutions are repeated if they share the genotype/the phenotype/part of them).

As for crossover operator, the authors explain in lines 210-212 that it is applied to swarm members only, but this sentence does not explain which specific operation is implemented to combine the positions of the swarms and how many swarm members are involved.

Experimental design

Methodological aspects have been reorganized and a new section for evaluation criteria has been added. However, details of the comparison algorithms are still missing, in particular: short description of all methods (not only MOC-GaPBK), justification of their choice and parameter values. This information is essential to ensure that the comparison was planned under fair conditions.

The authors reply that “The results have been computed for different number of runs” to my comment about the number of executions and statistical validation. Considering that the algorithms need at most 11h to run over the biggest dataset, I do not see any reason to have different number of runs at least for smaller datasets. Also, the authors should detail if the comparison algorithms are randomized or not, and how many runs were reported in all cases. The authors do not explain why statistical validation is not included. I mean null hypothesis testing [1] to check if MOWOATS wins across all datasets for which the same group of algorithms is executed.

[1] J. Derrac, S. García, D. Molina, F. Herrera. “A practical tutorial on the use of nonparametric statistical tests as a methodology for comparing evolutionary and swarm intelligence algorithms”. Swarm and Evolutionary Computation. 2011.

Validity of the findings

The authors reply that comparison algorithms were not run by themselves over the same platform used for the evaluation of MOWOATS. Also, that it is not possible because they do not have access to the server anymore. So, execution time is not directly comparable, of course. While I understand the technical problem with the server, it implies that the analysis is somehow biased and not as useful as it should be for this kind of study. The authors should clearly state that comparison is based on values reported by the original papers and revise sentences that suggest they execute all algorithms (e.g. line 370). If possible, they should collect information about the platform and execution time in the original papers and try to build some discussion upon that. Otherwise, the reader only knows part of the story (MOWOATS is better in the evaluation metrics), but nothing can be inferred about how much extra time is needed to achieve the improvements.

Additional comments

The authors have rephrased many parts of the manuscript to incorporate most of the requested comments. However, there are still some aspects that have not been properly addressed or need further clarification.

---

## Round 0.3 · accepted · Accept

Thank you for addressing the reviewers comments